

# Caffeoylquinic acid profiling: comparative analysis in yerba mate, Indian camphorweed, and stevia extracts with emphasis on the influence of brewing conditions and cold storage in yerba mate infusion

Gholamreza Khaksar[1], Nantachaporn Chaichana[2], Kitipong Assatarakul[2] and Supaart Sirikantaramas[1,3]

[1] Center of Excellence in Molecular Crop, Department of Biochemistry, Faculty of Science, Chulalongkorn University, Bangkok, Thailand
[2] Department of Food Technology, Faculty of Science, Chulalongkorn University, Bangkok, Thailand
[3] Omics Sciences and Bioinformatics Center, Chulalongkorn University, Bangkok, Thailand

Corresponding author
Supaart Sirikantaramas,
supaart.s@chula.ac.th

## ABSTRACT

Herbal infusions exhibit diverse pharmacological effects, such as antioxidant, anti-inflammatory, anticancer, antihypertensive, and antineurodegenerative activities, which can be attributed to the high content of phenolic compounds (*e.g.*, caffeoylquinic acids (CQAs)). In this study, we used ultraperformance liquid chromatography to determine the content of CQAs in the methanolic extracts of model herbs, namely, yerba mate (*Ilex paraguariensis*), stevia (*Stevia rebaudiana*), and Indian camphorweed (*Pluchea indica* (L.) Less.). The results revealed that yerba mate had the highest total CQA content (108.05 ± 1.12 mg/g of dry weight). Furthermore, we evaluated the effect of brewing conditions and storage at 4 °C under dark and light conditions on the antioxidant property and total phenolic and CQA contents of a yerba mate infusion. The analysis of the yerba mate infusions prepared with different steeping times, dried leaf weights, and water temperatures revealed that the amount of extracted CQAs was maximized (~175 mg/150 mL) when 6 g of dried leaves were steeped in hot water for 10 min. A total of 10-day refrigerated storage resulted in no significant changes in the antioxidant activity and total phenolic and CQA contents of an infusion kept in a brown container (dark). However, the antioxidant properties and total phenolic and CQA contents were negatively affected when kept in a clear container, suggesting the detrimental effect of light exposure. Our study provides practical recommendations for improving the preparation and storage of herbal infusions, thus catering to the needs of consumers, food scientists, and commercial producers. Moreover, it is the first study of the influence of light exposure on the content of crucial quality attributes within plant-based beverages.

## INTRODUCTION

As naturally occurring plant-derived substances, herbal medicines have long been used to treat numerous diseases and are mainly consumed in the form of herbal teas prepared as infusions (by steeping dried plant parts such as flowers, leaves, seeds, roots, and bark in hot water) or decoctions (by boiling herbs in water) (*Sentkowska, Biesaga & Pyrzynska, 2016*).

The health benefits provided by herbal teas are mainly attributed to their high contents of natural bioactive compounds such as phenolic compounds (*Shahrzad & Bitsch, 1996*; *Chandrasekara & Shahidi, 2018*). Caffeoylquinic acids (CQAs) are biologically important phenolics that are derived from hydroxycinnamic acid and exhibit numerous health-beneficial properties such as antioxidant, anti-inflammatory, anticancer, antidiabetic, antihypertensive, and antineurodegenerative activities (*Santana-Gálvez, Cisneros-Zevallos & Jacobo-Velázquez, 2017*). Based on the position, number, and identity of their acyl residues, these compounds, which are naturally produced by plants in response to biotic and abiotic stressors (*Farah & Donangelo, 2006*), can be categorized into three main groups, namely, monocaffeoylquinic acids (monoCQAs: 3-CQA, 4-CQA, and 5-CQA; known as chlorogenic acids), dicaffeoylquinic acids (diCQAs: 3,4-diCQA, 3,5-diCQA, and 4,5-diCQA), and feruloylquinic acids (FQAs: 3-FQA, 4-FQA, and 5-FQA), which have been identified in green coffee beans (*Clifford & Wight, 1976*) and green-coffee-containing food supplements (*Kremr et al., 2017*).

Previous studies have well documented the presence of CQAs in various herbal infusions. *Marques & Farah (2009)* identified and quantified CQAs in the infusions of 14 medicinal plants, with the highest CQA contents found in *Ilex paraguariensis* (yerba mate) infusion. Based on the fact that the CQA content of toasted yerba mate was much lower than that of green yerba mate, the authors concluded that CQAs are heat-sensitive. *Meinhart et al. (2018)* profiled the content of CQAs and caffeic acid in the infusions of 89 plants traditionally used and commercialized in Brazil, revealing that 93% of the tested infusions contained CQAs. The content of CQAs was highest for yerba mate (52.6 mg in 100 mL of infusion). *Chewchida & Vongsak (2019)* reported the presence of mono- (~2.17 wt% 5-CQA) and diCQAs (~1.24 wt% 3,4-diCQA and ~1.93 wt% 3,5-diCQA) in the infusion of Indian camphorweed (hereafter, pluchea) (*Pluchea indica* (L.) Less.), a traditional Thai medicine widely used because of its health-promoting properties. *Karaköse et al. (2015)* profiled the CQA content of stevia (*Stevia rebaudiana* Bertoni), a plant commonly used for sweetening, medicinal, pharmaceutical, and feeding purposes, showing that the corresponding leaf extract contained 5-CQA (~2.69 g/100 g of dry weight (DW)) and 4,5-diCQA (~1.68 g/100 g of DW).

The brewing conditions such as steeping temperature, steeping time, leaf particle size, and water/leaf mass ratio can significantly affect the antioxidant activity and total phenolic content of different kinds of teas, as verified by research of green, oolong, black, and fruit teas (*Khokhar & Magnusdottir, 2002*), green tea (*Komes et al., 2010*), black tea (Chinese Lapsang Souchong), white tea (Chinese Pai Mu Tan), green tea (China Special Gunpowder), oolong tea, and one blended black tea (Lyons Gold Brand, which is a blend of African and Indian black teas) (*Venditti et al., 2010*), green, oolong, and black tea

(*Yuann et al., 2015*), white, green, and black tea (*Hajiaghaalipour, Sanusi & Kanthimathi, 2016*), and black tea (*Chang et al., 2020*). However, the effects of brewing parameters on the antioxidant activity and phenolic content of herbal infusions remain underexplored and have only been examined by *Sentkowska, Biesaga & Pyrzynska (2016)*, who investigated the effect of steeping time on the antioxidant activity and phenolic content of chamomile (*Matricaria chamomilla* L.) and St. John's wort (*Hypericum perforatum*) infusions. Considering the numerous health benefits of CQAs, a deeper understanding of the effects brewing conditions have on the CQA profiles of herbal infusions is of upmost importance for consumers and food scientists.

Another important factor possibly impacting the antioxidant activity and total phenolic content of herbal infusions is low-temperature storage (4 °C). However, the data concerning the effect of low-temperature storage on the quality of teas and infusions, including the catechin content of green tea (*Labbé et al., 2008*; *Bazinet et al., 2010*; *Ananingsih, Sharma & Zhou, 2013*), antioxidant capacity, total phenolic content, and color analysis of 36 plants traditionally consumed in Spain as infusions (*Jiménez-Zamora, Delgado-Andrade & Rufián-Henares, 2016*), and antioxidant activity and total phenolic content of black tea (*Chang et al., 2020*), are inconsistent. Further, to the best of our knowledge, the effect of refrigerated storage on the CQA content of herbal infusions has not been explored. When investigating the effect of storage on CQA content, one should consider temperature and light exposure, as CQAs can be degraded or isomerized at high temperatures or upon irradiation with light (*Xue et al., 2016*). For example, during storage inside refrigerators, especially those with glass doors, beverages can be periodically exposed to light (*Ellie Flair Beverage Appliances, 2024*).

In this study, we determined the content of CQAs in the methanolic extracts of model herbs, namely, yerba mate, stevia, and pluchea, and prepared infusions of the herb with the highest CQA content (yerba mate) to investigate the effect of brewing conditions and low-temperature storage with and without light exposure on the CQA content. Thus, our findings provide health-conscious consumers with a better understanding of the optimal conditions for herbal infusion preparation and storage, while providing a first-time account of the potential effect of light exposure on the CQA content of a plant-based drink.

## MATERIALS AND METHODS

### Reagents

Portions of this text were previously published as part of a preprint (https://www.researchsquare.com/article/rs-2831137/v1). Commercial standards (monoCQAs: 1-CQA, 3-CQA, 4-CQA, and 5-CQA; diCQAs: 1,3-diCQA, 1,4-diCQA, 1,5-diCQA, 3,4-diCQA, and 4,5-diCQA) were purchased from Biosynth Carbosynth®, UK (purity ≥ 98.0%), and 3,5-diCQA was purchased from TransMIT GmbH PlantMetaChem (Gerlingen, Germany). Puerarin (internal standard) was obtained from Sigma-Aldrich, Inc., (St. Louis, MO, USA) (purity ≥ 98.0%). Acetonitrile and methanol (HPLC grade) were obtained from Merck (Darmstadt, Germany). Ultrapure water prepared using the Milli-Q system (Millipore, Burlington, MA, USA) was used in all experiments. All reagents used to determine antioxidant activity (1,1-diphenyl-2-picrylhydrazyl (DPPH)) and total phenolic
content (Folin–Ciocalteu reagent, gallic acid) were obtained from Sigma-Aldrich, Inc., (St. Louis, MO, USA) (purity ≥ 98.0%).

## Plant materials

Plant samples were purchased from reliable commercial sources (certified by the Thai Food and Drug Administration (FDA)) in Bangkok, Thailand in the form of dried leaves in sachets, which is how they are usually consumed. Three commercial brands of each plant species (yerba mate, stevia, and pluchea) herbal tea were obtained in sealed packages containing sachets. These brands are certified by the Thai FDA and use the same plant species and processing method (air drying). For stevia and pluchea, Thai commercial producers use the leaves from locally cultivated plants. However, for yerba mate, commercial producers import the leaves, mainly from South America. It should be noted that the plant species for each brand was verified and authenticated by a taxonomist and curator using the identification key provided in Flora of Thailand (*Haider, 2011*) (for stevia and Indian camphorweed) and by a botanist using molecular methods (for yerba mate) before the manufacturing process. This is a prerequisite for all commercial producers for the authentication of herbal tea products according to the Thai FDA. All samples were analyzed without further drying to maintain the original CQA content. Additional information regarding the plant samples used in this study is presented in Table S1.

## Sample extraction

Samples were ground to a fine powder using a mixer mill (MM 400, Retsch GmbH, Gerlingen, Germany) at 30 Hz for 1 min until they could pass through a 0.75 mm sieve (*Marques & Farah, 2009*). A 20-mg aliquot (DW) of each ground sample was extracted with 1 mL of 80% (v/v) aqueous methanol containing puerarin (0.05 g $L^{-1}$) as an internal standard upon vigorous 15-min shaking at 1,500 rpm and 15 °C (Eppendorf Thermomixer® C, Eppendorf, Hamburg, Germany). The resulting mixtures were centrifuged at 12,000 × $g$ and 4 °C for 15 min using a table-top centrifuge (5415 R, Eppendorf, Hamburg, Germany), and the supernatants were collected, filtered through a 0.2-µm nylon syringe filter, and injected into a high-performance liquid chromatograph for CQA analysis (*Cheevarungnapakul et al., 2019*; *Khaksar et al., 2021*).

## Infusion preparation

The effects of brewing conditions on the antioxidant activity, total phenolic content, and total CQA content were investigated for yerba mate, which had the highest CQA content. The optimal brewing conditions were determined through systematic investigation, and these conditions were employed to prepare infusions for subsequent experiments, as illustrated in Fig. 1.

## Effects of water/leaf mass ratio and steeping time

To investigate the effect of different water/leaf mass ratios, yerba mate infusions were prepared using varying amounts of dried leaves as follow: 2 g (equivalent to the plant mass in a commercial sachet), 4 g (two tea bags), 6 g (three tea bags), and 8 g (four tea bags). The tea bags were dipped into 150 mL (volume of a medium-size cup)

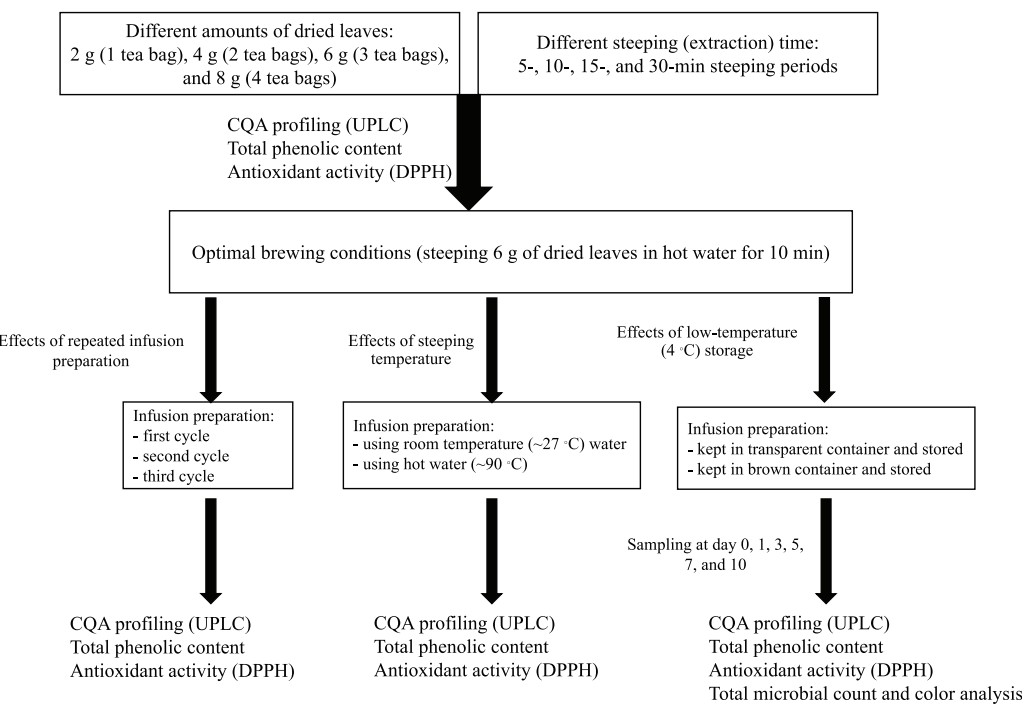

Figure 1 Schematic summarizing the infusion preparation procedures used in this study.

(*Jiménez-Zamora, Delgado-Andrade & Rufián-Henares, 2016*) of freshly boiled water (~90 °C) and steeped for 5 min (*Bastos et al., 2005*; *Pękal et al., 2011*). After infusion, the bags were removed and the liquid was immediately cooled in an ice bath (*Chang et al., 2020*) and filtered through paper (Whatman No. 1) (*Chewchida & Vongsak, 2019*). Then, the infusions were filtered through a 0.2 μm nylon syringe filter (*Cheevarungnapakul et al., 2019*) and analyzed (chromatography, total phenolic content, and antioxidant activity). To investigate the effect of steeping (extraction) time on the CQA content, different infusions were prepared using steeping periods of 10, 15, and 30 min (*Komes et al., 2010*). The corresponding infusions were then filtered and prepared for analysis as described previously. The identified optimal brewing conditions (steeping 6 g of dried leaves in hot water for 10 min) were used for subsequent experiments.

## Effects of repeated infusion preparation

Many consumers repeat the infusion preparation by adding water to steeped leaves. Accordingly, to examine the effect of this process on CQA content, a yerba mate infusion was prepared using hot water under the optimal conditions obtained previously. After filtering, the infusion preparation was repeated by adding 150 mL of hot water to the steeped leaves (second cycle), and this process was repeated one more time (third cycle) (*Komes et al., 2010*). After each cycle, the infusion was filtered through a 0.2 μm nylon syringe filter (*Cheevarungnapakul et al., 2019*) and analyzed (chromatography, total phenolic content, and antioxidant activity) as described in the related sections.

## Effects of steeping temperature

Steeping with cool water is a common practice among yerba mate consumers, especially in Paraguay and Southern Brazil (*Mateina, 2019*). Accordingly, to reproduce these conditions and investigate the effect of steeping temperature on the CQA content, total phenolic content, and antioxidant activity, we prepared an infusion using room temperature water (~27 °C) (*Venditti et al., 2010*; *Yuann et al., 2015*) under the optimal conditions obtained previously, subjected it to the corresponding analyses, and compared the results with those obtained for the infusion prepared using hot water (~90 °C).

## Effects of low-temperature storage

To investigate the effect of storage under simulated refrigeration conditions on the antioxidant activity and total phenolic and CQA contents, yerba mate infusion was prepared under the optimal conditions obtained previously. The infusion was then filtered and stored inside brown (dark conditions) and transparent (light conditions) air-tight glass containers inside a glass-doored refrigerator (4 °C) (*Labbé et al., 2008*; *Bazinet et al., 2010*; *Chang et al., 2020*) for 10 days. Control samples (day 0) were obtained before refrigerated storage. After 1, 3, 5, 7, and 10 days of storage, the samples were filtered through a 0.2 µm nylon syringe filter (*Cheevarungnapakul et al., 2019*) and analyzed.

## Antioxidant activity determination

The DPPH radical scavenging activity was evaluated using the methodology outlined by *Brand-Williams, Cuvelier & Berset (1995)*, with slight modifications as detailed by *Khaksar, Assatarakul & Sirikantaramas (2019)*. Briefly, the DPPH working solution was prepared by combining 10 mL of stock solution (24 mg of DPPH dissolved in 100 mL of methanol) with 45 mL of methanol. Subsequently, 950 µL of the DPPH solution was blended with 50 µL of the filtered infusion sample and incubated for 1 h in the dark. DPPH radical scavenging activity was calculated as:

DPPH radical scavenging activity (%) = $100 \times (A - A_1 + A_2)/A$,

where $A$ (absorbance) is the $A_{515}$ of the DPPH solution without the infusion, $A_1$ is the $A_{515}$ of the DPPH solution with the infusion, and $A_2$ is the $A_{515}$ of the infusion.

## Total phenolic content

Total phenolic content was measured according to the Folin-Ciocalteu method (*Swain & Hillis, 1959*). A 150 µL aliquot of each filtered infusion sample was mixed with 2,400 µL of ultrapure water and 150 µL of 0.5 N Folin–Ciocalteu reagent, thoroughly combined using a vortex mixer, and left to react for 30 min at room temperature (~28 °C). Following this, 300 µL of 1 N $Na_2CO_3$ solution was added, mixed well, and the mixture was incubated in the dark at room temperature for 2 h. The absorbance was then measured at 725 nm using a microplate reader (Tecan Infinite 200 PRO microplate reader). Total phenolic content was expressed as g gallic acid equivalent (GAE) per 150 mL.

## Total microbial count determination and color analysis

The total microbial count and color stability are important quality characteristics of beverages during storage (*Ashurst, 2016*). Total microbial count was determined following the method described in our previous study (*Khaksar, Assatarakul & Sirikantaramas, 2019*). The results were expressed as log colony-forming units (CFU) per mL of sample. For color analysis, three color parameters, namely $L^*$ (lightness), $a^*$ (redness/greenness), and $b^*$ (yellowness/blueness), were measured using a colorimeter (CR-410 chroma meter, Minolta, Tokyo, Japan). The color difference ($\Delta E$) was calculated (*Khaksar, Assatarakul & Sirikantaramas, 2019*) using the freshly prepared infusion as a control:
$\Delta E = [(\Delta L^*)^2 + (\Delta a^*)^2 + (\Delta b^*)^2]^{0.5}$.

## Ultraperformance liquid chromatography (UPLC)

The CQA profiling of infusions was performed using an UltiMate 3000 UPLC system coupled with a Dionex UltiMate DAD 3000 detector (Thermo Fisher Scientific, Waltham, MA, USA) and equipped with a Kinetex EVO C18 column (250 mm × 4.6 mm, 5 μm) (Phenomenex, Torrance, CA, USA) at a detection wavelength of 325 nm following the method described in our previous study (*Khaksar et al., 2021*). Peaks matching the retention times and UV spectra of commercial standards were identified as CQAs. Puerarin was used as an internal standard. Each CQA was quantified according to its calibration curve in the range of 1.30–500 μg/mL. More information on the employed CQA standards, including standard curves and equations, limit of detection, and limit of quantification, is presented in Fig. S1 and Table S2.

## Statistical analysis

The acquired data were subjected to statistical analysis using GraphPad Prism version 10.1.2 for Mac (GraphPad Software, Boston, Massachusetts USA) and are presented as the mean ± standard deviation of three independent replicates. The statistical comparisons of the means were carried out using one-way ANOVA followed by Tukey's honest significant difference (HSD) *post-hoc* test (*Midway et al., 2020*). Statistical significance was set at $p < 0.05$.

# RESULTS

## CQA profiles of methanolic extracts

Six CQAs (3-CQA, 4-CQA, 5- CQA, 3,4-diCQA, 3,5-diCQA, and 4,5-diCQA) were identified and quantified in the investigated plants (Table 1). The total CQA content decreased in the order of yerba mate (108.05 ± 1.12 mg/g DW basis) > stevia (79.26 ± 0.74 mg/g DW basis) > pluchea (64.59 ± 1.18 mg/g DW basis). Notably, whereas yerba mate and stevia contained all six of the above CQAs, pluchea contained only five (*i.e.*, did not contain 4-CQA). The contributions of mono- and diCQAs to the total CQA content depended on the plant, *i.e.*, the major contributor was identified as monoCQAs for yerba mate and stevia and as diCQAs for pluchea. The distribution of monoCQAs also depended

**Table 1 Caffeoylquinic acid (CQA) contents of methanolic plant extracts.**

| Sample | 3-CQA | 4-CQA | 5-CQA | 3,5-diCQA | 3,4-diCQA | 4,5-diCQA | Total CQA |
|---|---|---|---|---|---|---|---|
| *Ilex paraguariensis* (brand A) | 25.11 ± 0.97[a] | 18.11 ± 0.44[a] | 23.11 ± 0.44[a] | 21.22 ± 1.01[a] | 7.11 ± 0.55[b] | 12.22 ± 0.41[a] | 108.05 ± 1.12[a] |
| *Ilex paraguariensis* (brand B) | 21.69 ± 1.02[b] | 14.90 ± 1.01[b] | 18.88 ± 0.67[b] | 21.01 ± 0.37[a] | 7.53 ± 0.41[b] | 12.22 ± 0.85[a] | 98.21 ± 0.86[b] |
| *Ilex paraguariensis* (brand C) | 21.55 ± 0.97[b] | 15.02 ± 0.44[b] | 18.01 ± 1.11[b] | 20.35 ± 0.73[a] | 7.33 ± 0.55[b] | 12.01 ± 0.86[a] | 97.69 ± 0.77[b] |
| *Stevia rebaudiana* (brand A) | 16.66 ± 0.65[c] | 9.11 ± 0.60[c] | 19.61 ± 0.31[b] | 16.16 ± 0.54[b] | 6.43 ± 0.51[b] | 9.22 ± 0.24[b] | 78.79 ± 1.07[c] |
| *Stevia rebaudiana* (brand B) | 17.11 ± 0.43[c] | 9.04 ± 0.51[c] | 19.22 ± 0.31[b] | 16.79 ± 0.36[b] | 6.21 ± 0.61[b] | 9.55 ± 0.30[b] | 79.11 ± 0.75[c] |
| *Stevia rebaudiana* (brand C) | 17.94 ± 0.12[c] | 9.12 ± 0.66[c] | 19.11 ± 0.23[b] | 17.01 ± 0.91[b] | 6.11 ± 0.81[b] | 8.97 ± 0.34[b] | 79.26 ± 0.74[c] |
| *Pluchea indica* (L.) Less. (brand A) | 3.11 ± 0.82[d] | nd | 23.97 ± 0.24[a] | 17.22 ± 0.63[b] | 10.65 ± 0.51[a] | 6.07 ± 0.14[c] | 63.11 ± 1.32[d] |
| *Pluchea indica* (L.) Less. (brand B) | 3.71 ± 0.21[d] | nd | 23.16 ± 0.31[a] | 17.11 ± 0.69[b] | 11.01 ± 0.93[a] | 5.99 ± 0.21[c] | 62.66 ± 0.96[d] |
| *Pluchea indica* (L.) Less. (brand C) | 4.66 ± 0.80[d] | nd | 24.01 ± 0.83[a] | 17.65 ± 0.55[b] | 11.11 ± 0.95[a] | 5.89 ± 0.11[c] | 64.59 ± 1.18[d] |

Note:
Data are presented as the means ± standard deviations of three independent replicates and expressed in mg/g of DW. For each compound, values in the same column followed by different superscript letters are significantly different ($p < 0.05$). nd, not detected (detection limit = 1.09 μg/mL).

on the plant, *i.e.*, 3-CQA was the most abundant monoCQA in yerba mate, while 5-CQA was most abundant in stevia and pluchea. Regarding diCQAs, 3,5-diCQA was the major isomer in all samples. Fig. 2 presents the chromatograms of the analytical standards and the yerba mate sample.

For each herbal tea, we observed similar distributions of CQAs across different commercial brands. No significant variation in the contents of individual isomers was found for stevia and pluchea, whereas for yerba mate, monoCQA isomers were significantly more abundant in brand A than in brands B and C (Table 1). In addition, we did not observe significant differences in total CQA content between methanolic and aqueous extracts of yerba mate.

## Effects of brewing conditions
### Effects of water/leaf mass ratio and steeping time
For 5 min steeping time (*Bastos et al., 2005*; *Pękal et al., 2011*), the antioxidant activity (Fig. 3A), total phenolic content (Fig. 3B), and total CQA content (Fig. 3C) of the yerba mate infusions increased with the increase in dried leaf weight, peaking at a dried leaf weight of 8 g. However, for steeping times of 10, 15, and 30 min, these parameters were maximized at a dried leaf weight of 6 g and did not change when this weight increased to 8 g (Fig. 3). At a constant dried leaf weight, an increase in steeping time from 5 to 10 min significantly increased antioxidant activity, total phenolic content, and total CQA content, whereas a further steeping time extension to 15 and 30 min had no effect (Fig. 3). Thus, the optimum brewing conditions corresponded to a steeping time of 10 min and a dried leaf weight of 6 g.

### Effects of repeated infusion preparation
Repeated infusion preparation negatively affected antioxidant activity (Fig. 4A), total phenolic content (Fig. 4B), and CQA content (Fig. 4C), all of which were minimized after the third cycle. Thus, the infusion with the highest antioxidant ability, total phenolic content, and total CQA content was that prepared using a single steeping cycle.

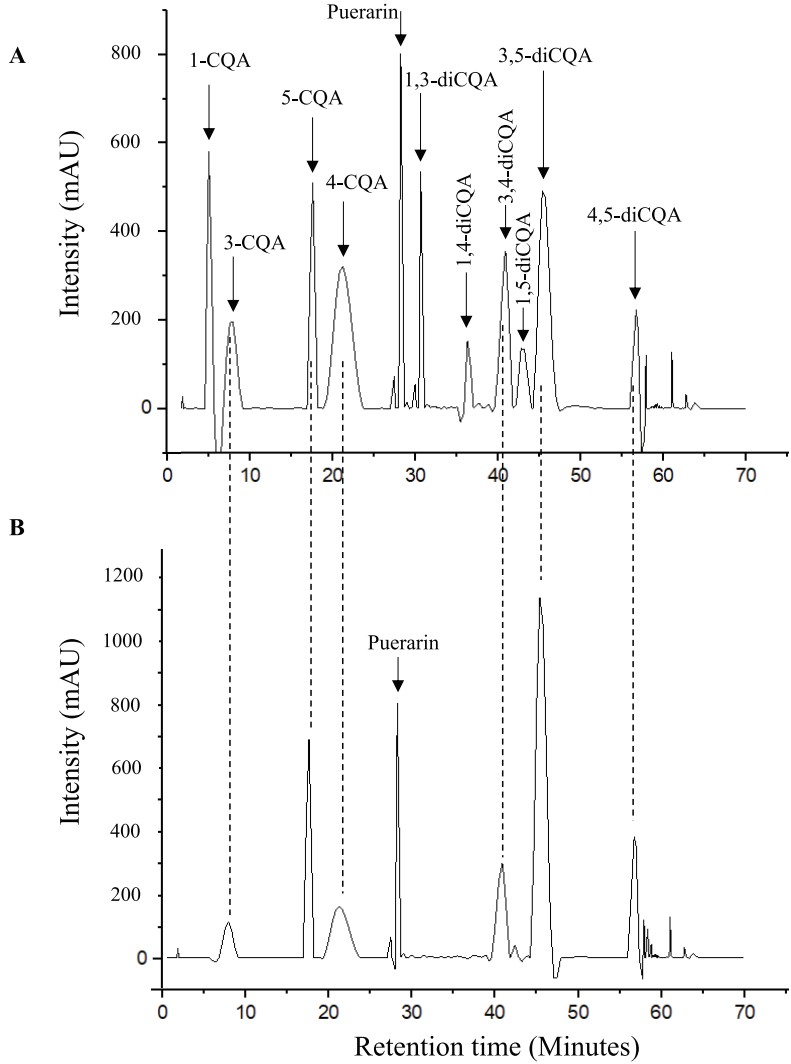

**Figure 2 Chromatograms of (A) analytical standards and (B) yerba mate sample.** Chromatograms created using OriginPro (OriginPro, Version 2022. OriginLab Corporation, Northampton, MA, USA).

### Effects of steeping temperature

Given that CQAs are heat sensitive, we investigated the effects of steeping temperature on the quality of the resulting infusion. Surprisingly, the antioxidant activity, total phenolic content, and total CQA content of the infusion prepared using cool water were significantly lower than those of the infusion prepared using hot water (Fig. 5).

## Effects of simulated refrigerated storage

### Effects on antioxidant activity, total phenolic content, and total CQA content

The refrigerated storage and on-demand consumption of home-made infusions is a regular practice among consumers. We found that the storage of the yerba mate infusion in a brown glass container (dark condition) at 4 °C did not affect its quality over 10 days of storage; that is, the antioxidant activity (Fig. 6A), total phenolic content (Fig. 6B), and total

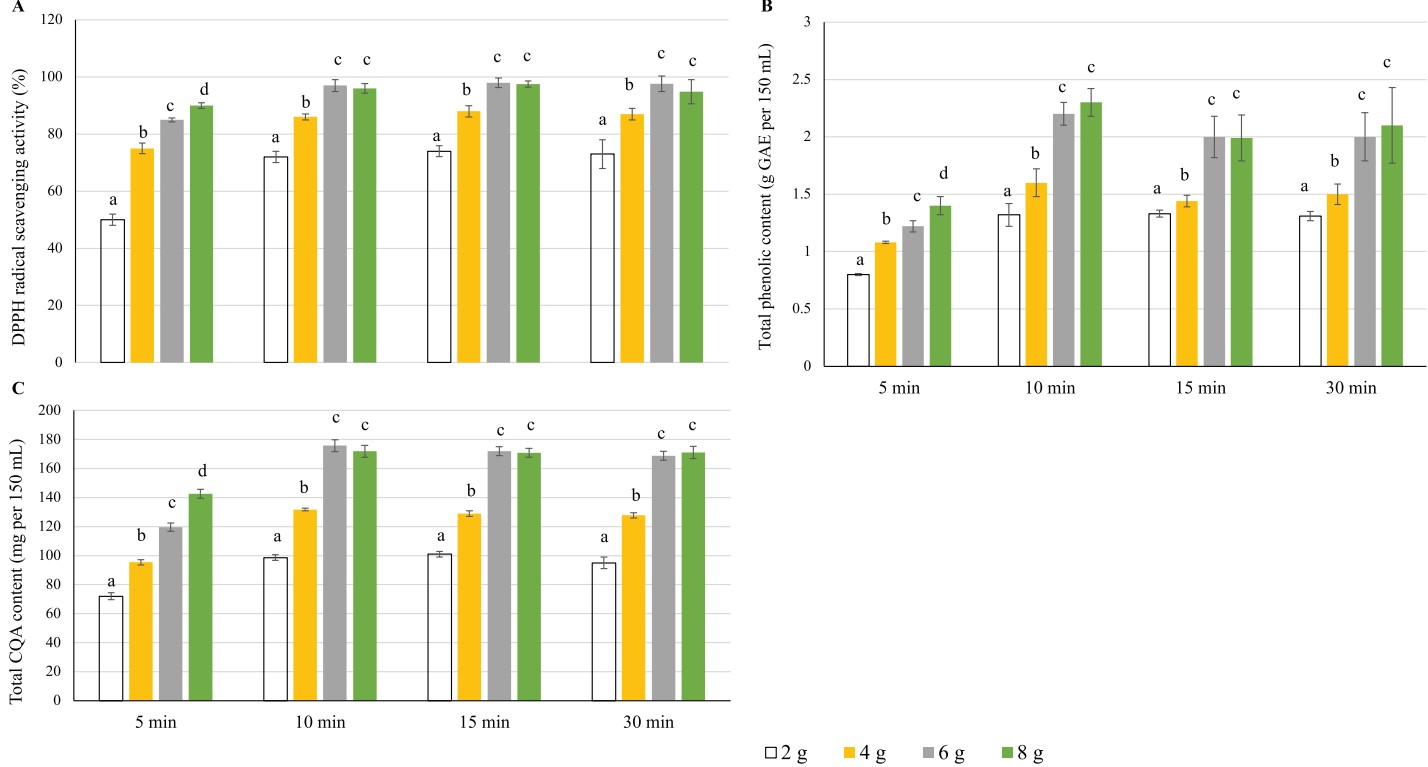

**Figure 3 Effects of the water/leaf mass ratio and steeping time on the (A) antioxidant activity, (B) total phenolic content, and (C) total CQA content of yerba mate infusions. Error bars represent mean ± standard deviation ($n$ = 3).** For each steeping time, comparisons are made among different leaf weights; bars with different letters differ significantly ($p < 0.05$).

CQA content (Fig. 6C) did not significantly deviate from those of the control sample (freshly prepared infusion). However, low-temperature storage negatively affected infusion quality when a transparent glass container (normal conditions) was used. As shown in Figs. 6A–6C, all measured parameters remained unchanged until day 7, significantly decreasing on day 10.

### Effects on total microbial count and color

Ten-day storage at 4 °C did not affect the total microbial count. During the storage period, the counts of total aerobic bacteria (Fig. S2A) and yeast/mold (Fig. S2B) did not significantly deviate from those of the control sample. Similarly, no significant changes were observed for $L^*$, $a^*$, and $b^*$ (Table S3). The color difference ($\Delta E$) did not significantly vary during storage, remaining at 0–1 ($0 < \Delta E < 1$), suggesting a non-observable difference (*Mokrzycki & Tatol, 2011*). Combining our results, which indicate no significant changes in the overall microbial (bacterial and yeast) count and color stability over the 10-day storage period at 4 °C, we can infer that the yerba mate infusion remained stable throughout storage.

### DISCUSSION

In this study, six CQAs, specifically 3-CQA, 4-CQA, 5-CQA, 3,4-diCQA, 3,5-diCQA, and 4,5-diCQA, were identified and quantified in methanolic extracts of dried leaves from

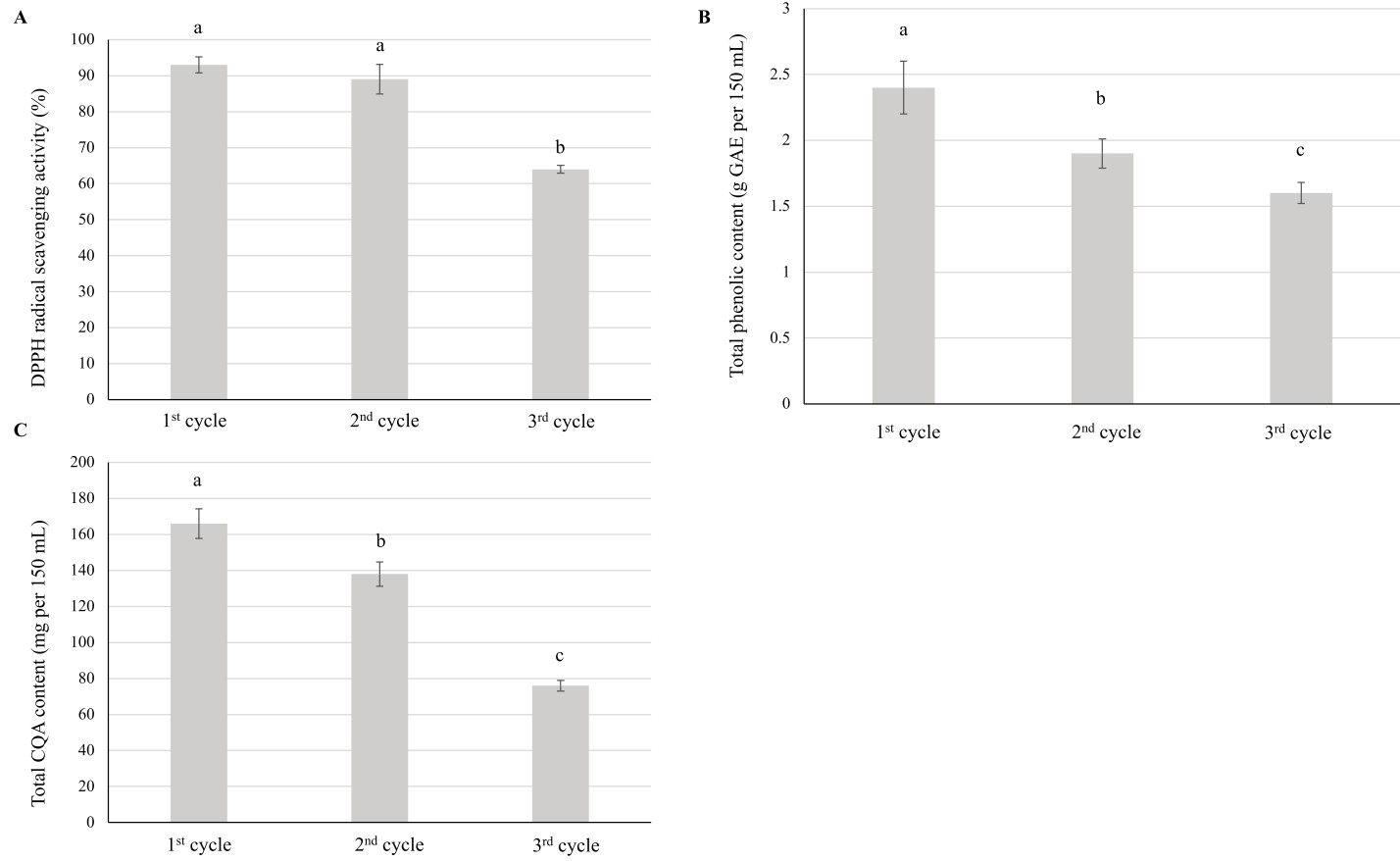

**Figure 4 Effect of repeated preparation on the (A) antioxidant activity, (B) total phenolic content, and (C) total CQA content of yerba mate infusions. Error bars represent mean ± standard deviation ($n = 3$).** For each parameter, comparisons are made between each of the three cycles; bars with different letters differ significantly ($p < 0.05$).

three model herbs: yerba mate, stevia, and pluchea. Yerba mate exhibited the highest total CQA content at 108.05 ± 1.12 mg/g of dry weight (Table 1). Notably, although yerba mate and stevia contained all six CQAs, pluchea lacked 4-CQA (Table 1).

One remarkable characteristic of CQAs is their propensity for structural modification through spontaneous acyl migration, specifically the migration of caffeoyl residues between the hydroxyl groups of the quinic acid moiety. Acyl migration contributes to the diversity of CQAs naturally present in plants, as previously reported (*Moglia et al., 2014*; *Xue et al., 2016*). The KNApSAcK database (http://www.knapsackfamily.com/KNApSAcK/) is valuable resource for comprehensively understanding the distribution of CQAs in plants. This database, which establishes species–metabolite relationships (*Afendi et al., 2012*), contains information on hundreds of plant species that produce CQAs. As of 2023, the database contains 169 plant species with 3-CQA, 54 with 4-CQA, and 15 with 5-CQA. This data underscores the prevalence and diversity of CQAs across various plant species.

The observed distributions of CQAs were notably similar across the various commercial brands of each herbal tea. No significant variation in the concentrations of individual

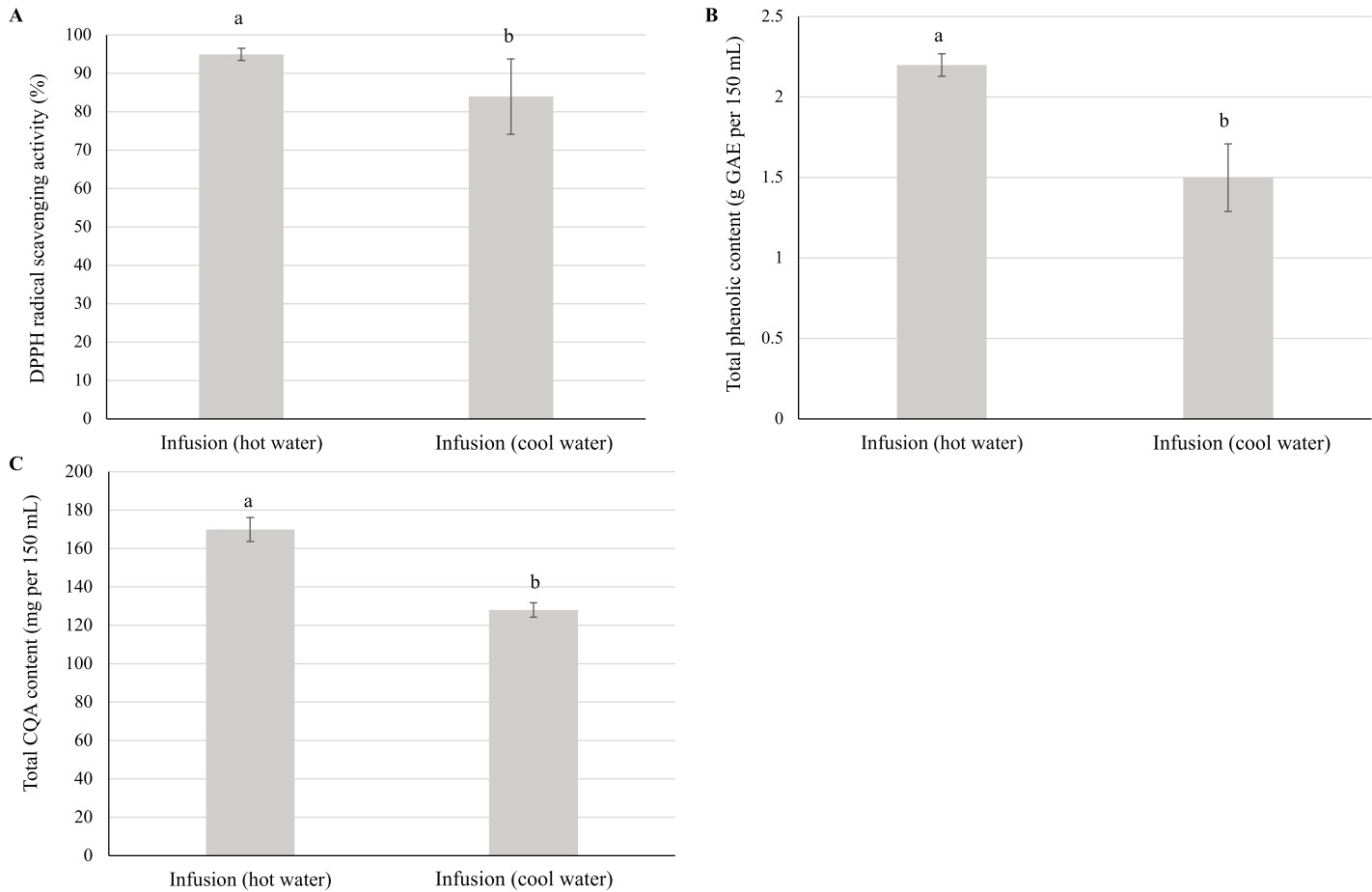

**Figure 5 Effect of steeping temperature on the antioxidant activity, total phenolic content, and total CQA content of yerba mate infusions.** Effect of steeping temperature (90 °C and 27 °C) on the (A) antioxidant activity, (B) total phenolic content, and (C) total CQA content of yerba mate infusions. Error bars represent mean ± standard deviation ($n = 3$). For each parameter, comparisons are made between infusions prepared using hot and cold water; bars with different letters differ significantly ($p < 0.05$).

isomers was detected for stevia and pluchea (Table 1). However, in the case of yerba mate, the monoCQA isomers were significantly more abundant in brand A than brands B and C (Table 1). Crucially, within the same plant species, the concentration of CQAs can be influenced by various factors, such as the plant's ontogenetic stage, genetics, biotic or abiotic stress, geography (cultivation and climatic conditions), and storage conditions (*Alcázar Magaña et al., 2021*). In this study, the dried leaves of stevia and pluchea were sourced from locally cultivated plants, thereby minimizing the potential effects of factors that could result in variations in the CQA content between different brands. However, for yerba mate, the dried leaves were imported from South America, suggesting likely differences in ontogenetic stage, genetics, biotic or abiotic stress, geography, and storage conditions in the different brands. Consistent with our findings, those of *da Silveira et al. (2017)* showed variations in the CQA contents among 15 commercial samples of yerba mate tea.

Yerba mate, widely consumed in South America, is well known for its high content of phenolics such as CQAs. A total of six CQA isomers (three monoCQAs and three diCQAs)

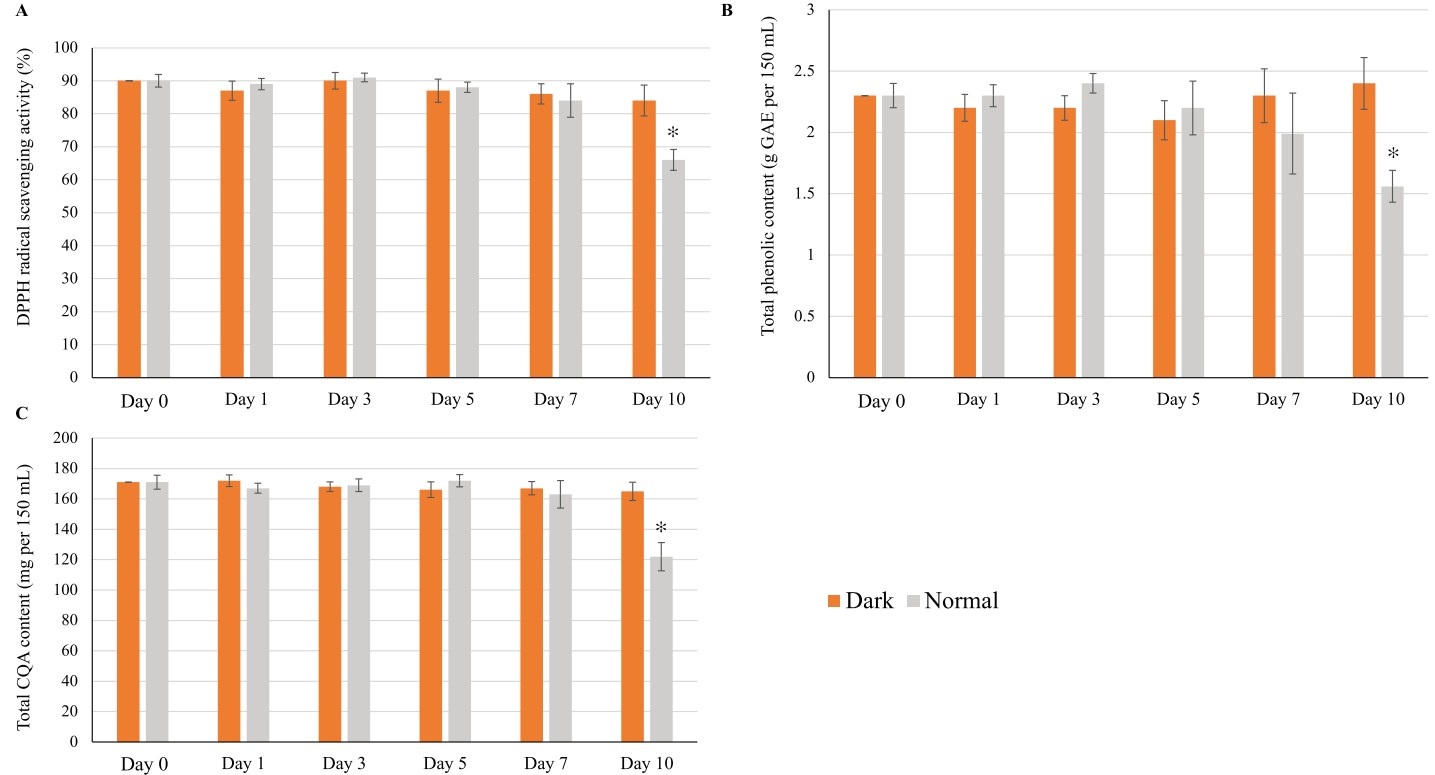

**Figure 6  Effect of low-temperature storage on the (A) antioxidant activity, (B) total phenolic content, and (C) total CQA content of yerba mate infusions.** Error bars represent mean ± standard deviations ($n = 3$). For each day, an asterisk (*) above the bars indicates a significant difference compared to the sample at day 0 (control) ($p < 0.05$). Full-size 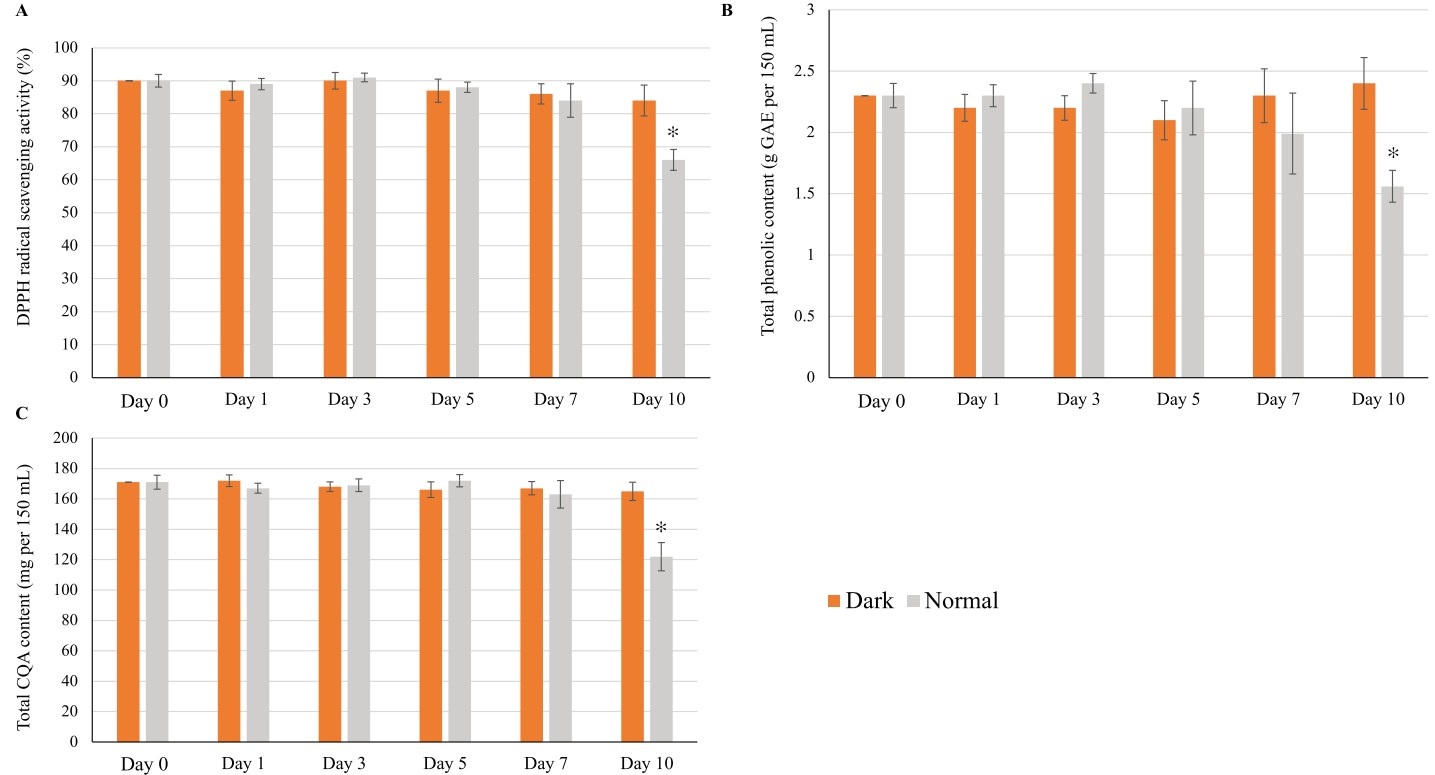 DOI: 10.7717/peerj.17250/fig-6

were identified and quantified in this plant. MonoCQAs accounted for 60% (on average) of the total CQA content and were mainly represented by 3-CQA, while diCQAs were mainly represented by 3,5-diCQA. Thus, our findings are in line with those of *Clifford & Ramirez-Martinez (1990)*, *Marques & Farah (2009)*, and *Meinhart et al. (2018)*. Notably, *Marques & Farah (2009)* reported the presence of caffeine in the methanolic extract of yerba mate (~15 mg per 100 g of DW), which is something caffeine-sensitive consumers should be aware of.

The methanolic and aqueous extracts of yerba mate had similar total CQA contents, in line with the report of *Marques & Farah (2009)*, which suggested that a satisfactory CQA extraction efficiency is achieved during the home preparation of yerba mate infusions. Data on the CQA content of beverages (except for coffee, which is well known to be rich in CQAs) are scarce. *Jeon et al. (2019)* determined the total content of CQAs in instant coffee (100%), ready-to-drink coffee (non-milk added coffee), and ground coffee (unblended and roasted) as ~80.7, 75.02, and 217 mg per 150 mL, respectively. A comparison of the cumulative CQA contents of the yerba mate infusion, which is ~175 mg per 150 mL, with that of coffee brews, highlights the CQA-rich nature of yerba mate infusions. Moreover, *Marques & Farah (2009)* determined the CQA content of some plant infusions commonly consumed in South America, including *Camellia sinensis* (green and black tea), *Melissa officinalis* (lemon balm), *Cymbopogon citratus* (lemon grass), *Cydonia oblonga* (quince),

and *Baccharis genistelloides* ("carqueja"). Notably, comparing the data on the total CQA content of these infusions, yerba mate contained a significantly higher total CQA content. *Shahrzad & Bitsch (1996)* reported cherry juice as a rich source of 5-CQA (chlorogenic acid), containing ~13 mg per 150 mL. Therefore, the 5-CQA content of yerba mate infusion (~11.5 mg per 150 mL) is comparable to that of the cherry juice. *Greenrod et al. (2005)* determined the total phenolic content of red wine (which is a good source of phenolics) as ~100 mg per 150 mL. Our data for the total phenolic content of yerba mate infusion revealed ~2 g GAE per 150 mL, also suggesting that yerba mate infusion is a rich source of phenolics.

The CQA-rich yerba mate commonly consumed as a homemade infusion deserves attention in view of the pharmacological activity of CQAs. However, not all plants having high CQA contents are good sources of CQAs for humans because the effects of the food matrix on CQA bioavailability remain unknown (*Rein et al., 2013*). Moreover, it would be incorrect to assert that the lower CQA contents of some herbal infusions are less important sources for humans because the metabolism and requirements of these compounds vary between individuals, and no dietary recommendations have been established (*Marques & Farah, 2009*).

During repeated steeping, CQA content was minimized at the end of the third cycle (~75 mg per 150 mL) (Fig. 4) but was still comparable to that of instant and ready-to-drink coffee, which suggested that satisfactory CQA extraction was achieved even upon repeated infusion preparation. Altogether, ~375 mg of CQAs (~58% of the total content in 6 g) was extracted over three cycles.

Herbal infusion preparation using cool water is a common practice among consumers, *e.g.*, infusions prepared by 2-h steeping at ~25 °C have become popular in Taiwan (*Venditti et al., 2010*). Herein, the antioxidant activity, total phenolic content, and total CQA content of the cold-brewed infusion were significantly lower than those of the hot-brewed infusion (Fig. 5), in agreement with the results previously obtained for the total phenolic content and antioxidant activity of teas prepared with cold water (*Khokhar & Magnusdottir, 2002*; *Venditti et al., 2010*; *Yuann et al., 2015*; *Hajiaghaalipour, Sanusi & Kanthimathi, 2016*). The effect of steeping temperature on CQA content observed herein was attributed to the increased solubility of CQAs in warm water (*De Maria et al., 1998*). Moreover, *Dibert, Cros & Andrieu (1989)* reported that the diffusivity of chlorogenic acids in water increases with increasing temperature. A rise in plant cell permeability with increasing temperature may also increase the transference rate of CQAs (*Dibert, Cros & Andrieu, 1989*). Notably, the preparation of the yerba mate infusion under optimum brewing conditions (steeping of 6 g of dried leaves for 10 min) using hot water did not have any detrimental effect on CQA content.

Although the refrigerated storage and on-demand consumption of homemade infusions is a regular practice, health-conscious consumers are often concerned about the effects of storage conditions on the nutritional qualities of infusions. Therefore, we investigated the effect of refrigerated storage on the antioxidant activity, total phenolic content, and CQA

content of a homemade yerba mate infusion over a 10-day period, revealing that these parameters were not affected in the case when storage was performed in a brown glass container but experienced deterioration when storage was performed in a transparent glass container (Fig. 6). Thus, our findings strongly suggest that exposure to light has a detrimental effect on the antioxidant activity, total phenolic content, and total CQA content of infusions. To the best of our knowledge, this is the first report on the effect of light on the CQA content of a drink.

Color stability and total microbial count are important quality characteristics of drinks during storage. Unlike in the case of CQA content, exposure to light did not affect the color stability and total microbial count of the infusion during storage (Table S3 and Fig. S2).

## CONCLUSIONS

The CQA contents of the methanolic extracts of three medicinal plants, namely, yerba mate, stevia, and Indian camphorweed (pluchea), were determined. Yerba mate had the highest total CQA content among the three plants. Recognizing the significant impact of brewing conditions on CQA extraction efficiency, we conducted an analysis of yerba mate infusions prepared using various brewing parameters. Optimal results, achieving ~175 mg per 150 mL of extracted CQAs, were achieved when 6 g of dried yerba mate leaves were steeped in hot water for 10 min. Additionally, we explored the impact of 10-day refrigerated storage on the antioxidant activity and total phenolic and CQA contents of yerba mate infusions. The results indicate that storage in a brown container maintained these properties, resulting in no significant changes, whereas storage in a clear container negatively affected the antioxidant properties and total phenolic and CQA contents. This observation highlights the detrimental impact of light exposure on yerba mate infusions. This study is the first to provide insights into the influence of brewing conditions and cold storage on the CQA content of yerba mate infusion. Furthermore, it contributes novel findings regarding the potential effect of light exposure on the CQA content of a plant-based drink.

## ACKNOWLEDGEMENTS

We thank Ms. Rawisada Pholsin and Ms. Pornpassorn Chulalaksananukul for their assistance with color and total microbial count analyses. We would like to thank Editage for English language editing.

### Funding

This research was supported by the Center of Excellence in Molecular Crop (to Supaart Sirikantaramas) and the Ratchadapisek Somphot Fund for Postdoctoral Fellowship (to Gholamreza Khaksar), Chulalongkorn University. The funders had no role in study design, data collection and analysis, decision to publish, or preparation of the manuscript.

## Grant Disclosures

The following grant information was disclosed by the authors:

Center of Excellence in Molecular Crop.

Ratchadapisek Somphot Fund for Postdoctoral Fellowship, Chulalongkorn University.

## Competing Interests

The authors declare that they have no competing interests.

## Author Contributions

- Gholamreza Khaksar performed the experiments, analyzed the data, prepared figures and/or tables, authored or reviewed drafts of the article, and approved the final draft.
- Nantachaporn Chaichana performed the experiments, prepared figures and/or tables, and approved the final draft.
- Kitipong Assatarakul conceived and designed the experiments, authored or reviewed drafts of the article, and approved the final draft.
- Supaart Sirikantaramas conceived and designed the experiments, analyzed the data, authored or reviewed drafts of the article, and approved the final draft.

## Data Availability

The data including standard curve, LOD/LOQ data and additional data are available in the Supplemental Files.

## Supplemental Information

Supplemental information for this article can be found online at http://dx.doi.org/10.7717/peerj.17250#supplemental-information.

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
