# Peer review of "Caffeoylquinic acid profiling: comparative analysis in yerba mate, Indian camphorweed, and stevia extracts with emphasis on the influence of brewing conditions and cold storage in yerba mate infusion"

_PeerJ, doi:10.7717/peerj.17250_

## Round 0.1 · original submission · Major Revisions

All three reviewers agreed that the paper is interesting and availability of the raw data makes it possible for the readers to reproduce the results. However, several important issues need addressing, and the lack of methodology description and referencing is the most important aspect of the critique.

1. Need to include all information related to sampling and sample characteristics.
2. Statistical methods need more careful description and justification. Please address the comments from Reviewer #3.
3. All methods need to have supporting references, please check the comments from Reviewers #1 and #2.
4. Reviewer #1 has a detailed list of suggestions for specific lines of the manuscript. I think that they can be very helpful in guiding the revision of your manuscript.
5. Please elaborate on the methods or protocols employed to ensure that the plants acquired for the study accurately corresponded to the specified species.
6. Please use the comments in the annotated review from Reviewer #2.

I hope that you find the comments useful and will be able to submit the revision in 35 days.

**Language Note:** PeerJ staff have identified that the English language needs to be improved. When you prepare your next revision, please either (i) have a colleague who is proficient in English and familiar with the subject matter review your manuscript, or (ii) contact a professional editing service to review your manuscript. PeerJ can provide language editing services - you can contact us at [email protected] for pricing (be sure to provide your manuscript number and title). – PeerJ Staff

·

Basic reporting

Pre-print available: https://doi.org/10.21203/rs.3.rs-2831137/v1

It is essential to ensure that all raw data supporting your conclusions are readily accessible for review and validation purposes (via Zenodo). I strongly encourage you to format raw data for clarity and ease of understanding.

This article provides the background and context of its field. The Introduction section demonstrates how the work fits into a broader field of knowledge. However, as noted in the additional comments section, there are a few areas where further improvements or clarifications could enhance the overall quality and comprehensibility of the manuscript.

This manuscript aligns with the journal’s aim and scope. The research question was defined as being relevant and meaningful. The manuscript identifies and articulates how this research fills a specific and relevant knowledge gap. Further specific recommendations for enhancement can be found in the additional comments.

Experimental design

To verify the plant species used and the consistency of their quality, it is crucial to address the identification of yerba mate, stevia, and Pluchea indica. Please elaborate on the methods or protocols employed to ensure that the plants acquired for the study accurately corresponded to the specified species. This is particularly important in phytochemical research, in which species identity can significantly affect the results. Additionally, please provide information on whether there are established quality criteria or regulatory standards for these herbal products in the country.

When reviewing the methodology section of your manuscript (lines 176-211), it is crucial to ensure that all methods and procedures are appropriately referenced, especially if they are adapted or derived from previous studies or standard practices in the field. Please review this section and confirm whether there are existing studies or standard methodologies that have informed your methods. If such references exist, incorporating them into the manuscript will significantly strengthen the methodological framework.

Validity of the findings

The manuscript has been carefully reviewed, and while it presents valuable research, there are areas that require further attention to align with the journal standards. This manuscript would benefit from a more detailed explanation of the methods used, particularly in terms of data collection and analysis. The conclusions of this study require further development to reflect the research conducted more accurately. Ensure that they are directly drawn from your results and are closely linked to your original research question.

Additional comments

• Lines 1-2 (Title): A title revision could enhance the clarity and comprehensiveness of the manuscript. Significant attention has been given to comparing yerba mate with stevia and Pluchea indica, which is not currently reflected in the title.

• Line 42- (abstract): The paragraph successfully introduces an interesting aspect of herbal infusions. Expanding the types of pharmacological effects would greatly enhance the abstract's informational value.

• Line 51: The phrase "above parameters" is somewhat vague and could lead to confusion. This would enhance clarity to explicitly state which parameters are being referred to, even if they have been mentioned in the previous sentence.

• Line 153: List the specific reagents used, including their chemical names and relevant concentrations or grades. If relevant, include the catalog numbers of the reagents. This would be particularly helpful for researchers looking to replicate the study and compare its results.

• Line 158: Clarify what makes these sources "reliable.” For instance, are they certified by any regulatory body, do they have a history of providing high-quality plant samples, or are specific quality checks performed?

• Line 159: It is essential to specify what these brands represent. Are there different commercial suppliers, cultivars, or varieties of the plant, or do they refer to different processing methods? Clarify how these brands were selected for the study. Was there a specific criterion or a set of criteria used? Examining multiple brands is a strength, as it introduces a comparative aspect to the study, potentially offering insights.

• Line 172: Providing detailed infusion preparation instructions is a strength, as it ensures that the methodology can be accurately replicated. Including a diagram or schematic to explain infusion preparation procedures would be highly beneficial, especially given the complexity of the process. Diagrams can aid in visually conveying the steps.

• Line 179: The reference to “most manufacturers” is somewhat vague. It would enhance the manuscript's reliability to cite specific sources or studies that support this standard steeping time.

• Line 215: The mention of modifications suggests an effort to refine or adapt the method to the specific context of the study, which is commendable if explained adequately. The modifications made, as mentioned in the manuscript, were entirely consistent with those described by Khaksar et al. 2019?

• Line 261: The observation that P. indica lacks 4-CQA is an intriguing finding that contributes to a broader understanding of these plants' characteristics. It may be beneficial to briefly discuss why the absence of 4-CQA in P. indica is noteworthy.

• Lines 269-274: Highlighting the significant difference in the content of monoCQA isomers in yerba mate across brands is a notable finding, adding depth to the understanding of CQA distribution in commercial products. More context or discussion regarding significance would enrich this analysis.

• Line 279: The paragraph references steeping time recommendations from "most manufacturers" without citing specific sources. It is important to have sources for such claims. Are the recommendations, as mentioned in the manuscript, available online or in publicly accessible documentation?

• Line 313: The paragraph presents the findings but does not explicitly state the broader purpose or relevance of these observations. Adding a brief statement explaining the significance of these findings in the context of the overall study would enhance the clarity.

• Lines 334-337: The paragraph currently juxtaposes data from different studies without a clear narrative or explanation of the purpose of the comparison. Providing this context would make the paragraph more informative and cohesive.

• Lines 341-344: This section addresses an important aspect of nutrition science. Presents an important perspective but lacks the empirical evidence necessary to support such claims. Including references to peer-reviewed studies strengthens this argument.

• Line 378: Make a stronger statement about the importance of brewing conditions and assert the novelty of the findings (because they are!).

·

Basic reporting

The theme is interesting; however, there are fundamental issues in the methodology that need clarification to support the results. Therefore, it is necessary to include all information related to sampling and sample characteristics to give meaning to the findings. The writing is clear, but it is important to update the references and add citations in the specified paragraphs. The conclusion needs reinforcement.

Experimental design

The number of plant materials, their sources, and the brands are not specified. How are these leaves sold? In sealed packages? Are they exposed to the environment? How are these leaves stored in stores? Do all acquired leaves have the same harvest or expiration date? These are fundamental data to verify the validity of the results that could influence the content of compounds present in the leaves. Did the researchers confirm that the purchased plants were the species indicated in their manuscript? Was this corroborated by a botanist or through molecular methods?

Validity of the findings

Points related to handling and processing are mentioned but not described in the methodology, and inferences about the results are made regarding these points. The authors refer to plants, but how can they confirm that each package, if collected in packages, comes from a single plant? Wouldn't it be more appropriate to talk about samples?

Additional comments

The manuscript is entirely available as a preprint at: https://www.researchsquare.com/article/rs-2831137/v1, as indicated by the authors on the journal platform. The title is appropriate and informative; however, the manuscript consists of two sections. In the first, three plants are studied, and then the study continues only with yerba mate, which is not reflected in the title. Two repetitions from the title should be replaced.

It partially reflects the content of the article, but there is room for improvement. The common name of Pluchea indica should be included. None of the methodology is present in the abstract and should be mentioned. References should be added in the indicated sections of the manuscript. It is advisable to add updated references. In paragraphs, full-text is written, and at the end, several citations are mentioned without specifying the points they refer to. It is recommended that each point have its specific reference.

Example:
"Brewing conditions such as steeping temperature, steeping time, leaf particle size, and water/leaf mass ratio can significantly affect the antioxidant activity and total phenolic content of black and green teas (Khokhar and Magnusdottir, 2002; Komes et al., 2010; Venditti et al., 2010; Yuaan et al., 2015; Hajiaghaalipour et al., 2016; Chang et al., 2020)."

Effects of water/leaf mass ratio and steeping time: This methodology lacks reference.
Effects of repeated infusion preparation: This methodology lacks reference.
Effects of steeping temperature: This methodology lacks reference.
Effects of low-temperature storage: This methodology lacks reference.
In Figure 1, mention the program used for its creation. In Figures 2, 3, 4, and 5, the black bar and deviation bar cannot be visualized correctly at the bottom due to the black color.

The structure is orderly and clear; however, it would be beneficial to delve deeper. Although the lack of bibliography is understandable, comparisons could be made with other food products (other herbs, fruits). Add the references where indicated.

The conclusion should be improved by directly addressing the results found in relation to the objectives, eliminating aspects related to the methodology. It is also not advisable to mention aspects related to health benefits since the authors have not studied this.

Reviewer 3 ·

Basic reporting

The manuscript is well-written and easy to understand. The research gap is well defined within the introduction.

Experimental design

The primary research is in the scope of the journal. The research question is also well-defined: the exact effects of different infusion preparation methods were not reported before. The authors studied the effects of infusion preparation parameters on the contents of the resulting solutions, specifically testing the antioxidant activity, phenolic contents, and total CQA contents.
The statistical analysis of differences in means between the groups was done using ANOVA followed by Duncan MRT or Student's t-test (lines 249-253). In my opinion, three replicates might not be enough for statistical testing like a t-test, and also it will be hard to reproduce the data.
The usage of Duncan MRT is not recommended based on inadequate error rate adjustment (specifically type I errors) (Kaller et al, 2020 (https://doi.org/10.7717%2Fpeerj.10387)). It would be valuable to reason the choice of this method.

Validity of the findings

The raw data and all the measurements on all the machines were provided together with the manuscript. The conclusions are well-stated and are supporting the results. The optimal conditions for the best extracts content-wise were determined and statistically tested (however, the usage of Duncan MRT does not seem like the best option to me).

---

## Round 0.2 · accepted · Accept

The editors were pleased with the changes, noting that the authors had met and exceeded the expectations, significantly improving the manuscript. I agree with their assessment and consider the text to be ready for publication.

·

Basic reporting

I have no further comments.--

Experimental design

I have no further comments.--

Validity of the findings

I have no further comments.--

Additional comments

Thank you sincerely for your resubmission of your manuscript. This meticulous study underscores its value as a substantial contribution to the field. It is evident that your work has the potential to influence future studies and spark meaningful discussion. With your thorough revisions, the manuscript not only meets, but also exceeds, the expected standards for publication. I am excited about the prospects of your work, reaching a wider audience and contributing to ongoing scholarly conversations.

·

Basic reporting

The authors have made the requested changes, the manuscript has been significantly improved, points have been clarified and missing references have been added

Experimental design

The requested points regarding sampling, statistical analysis and verification of the studied species have been clarified.

Validity of the findings

With the points clarified, the findings presented in this manuscript are strengthened and it is possible to understand them adequately.

Additional comments

no comment

Reviewer 3 ·

Basic reporting

no comment

Experimental design

no comment

Validity of the findings

no comment

Additional comments

Thank you for your work!